# Delineation of Orchard, Vineyard, and Olive Trees Based on Phenology Metrics Derived from Time Series of Sentinel-2

**Mukhtar Adamu Abubakar** [1,2], **André Chanzy** [1,*], **Fabrice Flamain** [1], **Guillaume Pouget** [1]
**and Dominique Courault** [1]

1    1114 UMR INRAE-Avignon University EMMAH, Domaine St. Paul, 84914 Avignon, France;
     mukhtar.abubakar@inrae.fr or dattijoh@nsuk.edu.ng (M.A.A.)
2    Agronomy Department, Faculty of Agriculture, Shabu-Lafia Campus, Nasarawa State University,
     Keffi 961101, Nigeria
*    Correspondence: andre.chanzy@inrae.fr

**Abstract:** This study aimed to propose an accurate and cost-effective analytical approach for the delineation of fruit trees in orchards, vineyards, and olive groves in Southern France, considering two locations. A classification based on phenology metrics (PM) derived from the Sentinel-2 time series was developed to perform the classification. The PM were computed by fitting a double logistic model on temporal profiles of vegetation indices to delineate orchard and vineyard classes. The generated PM were introduced into a random forest (RF) algorithm for classification. The method was tested on different vegetation indices, with the best results obtained with the leaf area index. To delineate the olive class, the temporal features of the green chlorophyll vegetation index were found to be the most appropriate. Obtained overall accuracies ranged from 89–96% and a Kappa of 0.86–0.95 (2016–2021), respectively. These accuracies are much better than applying the RF algorithm to the LAI time series, which led to a Kappa ranging between 0.3 and 0.52 and demonstrates the interest in using phenological traits rather than the raw time series of the remote sensing data. The method can be well reproduced from one year to another. This is an interesting feature to reduce the burden of collecting ground-truth information. If the method is generic, it needs to be calibrated in given areas as soon as a phenology shift is expected.

**Keywords:** woody crop classification; Sentinel-2; random forest; crop phenology; olive; orchard; vineyards; Mediterranean





## 1. Introduction

Among the several hazards of climate change and global warming on natural resources, the most significant threat is its implication for the accessible availability of freshwater. Unequivocally, the agricultural sector is the highest consumer of water worldwide, with irrigation accounting for about 70% of freshwater withdrawals [1–3]. Thus, supervision of irrigation activities is crucial to buttress the execution of water management policies and improve water use productivity [4,5]. Supervision of irrigation activities not only encompasses spatial assessments of areas under irrigation but also irrigation strategies [4–7], which differ between crop systems [8]. Therefore, mapping the different irrigated crops is an important issue in water management, particularly in the Mediterranean region, which is sensitive to variations in agricultural activities and land use due to its exposure to excessive climatic threats [9]. Irrigation patterns and water quantity depend on crop type and associated irrigation methods; for instance, flooding irrigation applied to grassland mobilizes a great quantity of water, while drip irrigation applied in horticultural production leads to frequent water supplies but with much less water. If numerous works address irrigated crop delineation, less attention has been paid to the delineation of perennial woody crops such as fruit trees in orchards, vineyards, and olive groves that are common in the Mediterranean.

Crop classification from remote sensing data is a field that has been widely studied for decades and is gaining interest with new satellite missions such as the Sentinel missions that have considerably improved temporal resolution and spectral richness. Progress in the identification of grasslands and field crops is undeniable [9–11]. On the other hand, the case of woody perennial crops such as fruit orchards, vineyards, or olive groves might pose more problems, but progress is still possible. The difficulty comes mainly from the fact that these covers have a great diversity of development because of the age of the plantation, their density, the mode of management such as pruning, and the confusion that there can be with other plant covers (non-irrigated meadows, wetlands, etc.).

Concerning woody crops, high-resolution Landsat TM images were used to identify crop classes (olive and citrus) in Marrakech, Morocco, using the temporal profile of the normalized difference vegetation index (NDVI) simply by setting a threshold of maximum and minimum values of the NDVI across the season [12] leading to an overall accuracy (OA) of 83%. Peña et al. [13] classified fruit trees by comparing Landsat 8 image times series considering the full band, the normalized difference water index (NDWI), and the normalized difference vegetation index (NDVI). The best results were obtained using the full spectral information, in particular with the visible and SWIRS bands (OA = 94%), while the NDVI led to the worst results. They tested the interest in dates and highlighted that the beginning (greenness) and end (senescence) of the growing cycle were the most significant phases for the separation. They obtained an OA of 94% with four dates. The interest in image acquisition during the greenness period was confirmed in [14]. In this study, it was demonstrated that up to seven types of orchards can be classified by considering all Landsat 8 spectral bands as well as a combination of bands. Recently, tree fruit crop type mapping was conducted in Egypt by examining various temporal windows, spectral approaches, and several combination methods between S1 (Sentinel-1) and S2 (Sentinel-2) data inserted into RF [15]. Good accuracy was found with S2 alone, while improvement was found by combining the textural S1 information with the spectral S2 observations, which led to an OA of 96%. In [16], a classification was carried out to delineate apple orchards, vineyards, and annual crops in Iran. Phenology was used to select the optimal dates. By combining S1, Landsat 8 images, and a digital elevation model, an OA of 89% was obtained. Another recent study was conducted in Juybar, Iran, where an automatic approach to mapping citrus orchards was implemented using S1 and S2 and the ALOS digital surface model (DSM) [17]. Without training and by considering a very large number of images (148), textural, and spectral features, it was possible to separate citrus and non-citrus surfaces with an OA of 99.7%. The context was very favorable with evergreen trees (citrus), which present a contrast with the other surfaces. These studies have shown that good results can be obtained with perennial woody crop mapping. The quality of the results obtained came from the number of images used, the choice of dates considered, and the complementarity between spectral indices in the optical domain and textural indices derived from SAR images. The quality of the classifications also came from the specificity of the signatures of the various covers. In this respect, the phenology makes it possible to target the dates of observation to be considered, in particular during the phases of greenness and senescence. In past studies, phenology was not used directly as a classification criterion but rather to determine optimal dates. The use of phenological traits may present advantages in the exploitation of time series due to the fact that they are relatively independent of the dates of acquisition. This can be interesting in a situation where partial cloud cover is frequent in the temperate zone and can disturb the homogeneity of the time series from one point to another in the area to be mapped. This can considerably disturb the learning algorithms.

Conventional crop phenology, also termed ground phenology (GP) [18], is the particular re-occurring events of crop life traits such as budburst, leaf development, senescence, flowering, and maturity [19], which is laborious to collect, time-consuming, and expensive as well [18,20]. These GP observations correlate to key plant physiological activities that govern natural resource uptake by plants. Despite GP remaining objective and precise, its characterization over a wide-scale area remains a challenge [21]. Satellite remote sensing

is capable of offering time series on vegetation development with a short revisit period, which can serve as a source of data to monitor vegetation phenology at a local and regional scale with proxies termed land surface phenology (LSP) [22]. Phenology metrics (PM) obtained from the analysis of vegetation index time series were often used to characterize the LSP [23–25]. In the past, most of the studies related to crop phenology were conducted using medium-resolution sensors (MODIS, AVHRR), allowing frequent acquisition over the whole globe [18]. The spatiotemporal resolution was enhanced by combining those medium-resolution sensors with high-resolution (LANDSAT) sensors [26,27]. Most research on LSP carried out using information from these satellite sensors is faced with the drawback of mixed pixels and is thus restricted in its implementation across complex or fragmented terrains [28]. Such a drawback can now be overcome by using S2, which allows accurate supervision of crop changes [29]. PM are linked to the variation of the seasonal pattern in cropland surfaces derived from satellite observations [30]. The most common patterns are the start of the season (SOS), the peak of the growing season (POS), the end of the season corresponding to senescence (EOS), and the length of the season (LOS) [30,31]. In other terms, in a growing year season, the major phases of phenology controlling the spectral patterns of vegetation are (i) the date of photosynthetic commencement (green-up), (ii) the date of maximum plant green leaf (maturity), and (iii) the date of decline in photosynthetic activities (senescence) [32]. The PM mentioned are normally computed from the common normalized difference vegetation index (NDVI) or other popular indices, for instance [31,33]. But despite that, the NDVI method can have some drawbacks, such as restricted sensitivity to vegetation photosynthetic dynamics [34], while biophysical variables such as LAI (leaf area index) can improve the PM, particularly for farmlands. The use of phenology as a classifier for crop mapping has been applied in many studies. In [35], PM (SOS, EOS, LOS, and the peak integral reflecting the photosynthetic activity) were derived from Modis NDVI time series using the TIMESAT algorithm [36] and used to characterize different agricultural systems (fallow, rainfed crop, irrigated crop, and irrigated perennial). It was shown that the PM were able to monitor agricultural system evolution across two decades, 2000–2019, with an OA ranging from 93% to 97%. In that case, irrigated perennials were evergreen orchards (citrus), which makes the distinction with annual crops easier. According to [37], they developed a phenology-based approach to delineate wheat and barley by identifying the heading date using the temporal features of the different S2 bands. Good results were obtained (OA of 76%) across three sites in Iran and the USA (North California and Idaho). These studies, among others, have shown that the PM can be used as a classifier to map crops. The quality of the results depends very much on the specificity of the temporal signatures of the different crops to be identified and the diversity of plant cover that can be found in a given class. Moreover, the added value of using PM rather than time series of spectral and/or vegetation index data has not yet been demonstrated.

The objective of this study is to characterize the main classes of perennial woody crops, namely fruit orchards (OC), vineyards (VY), and olive groves (OL), which are cropping systems with different irrigation strategies. Within these classes, there is a great diversity of situations marked by the type of cover, pruning practices, or soil management in the inter-row. To address this diversity of situations, we intend to rely on phenological traits to identify the crops studied in this work. Such approaches have proven to be successful in the identification of annual crops, and we assumed that such approaches could be interesting for perennial woody crops. Indeed, we believed that if the diversity of the characteristics of a crop type due to their management and their ages can lead to variable remote sensing signatures, these crops share the same phenological traits. The study is carried out on two sites about 100 km apart but with different climatic conditions and plant cover other than the desired perennial woody crops. The challenge will be to evaluate the performance of classifications carried out with PM, to analyze their adding value in comparison with approaches based on the time series of vegetation index, and to establish the genericity of a classification model from one year to another or from one site to another.

## 2. Materials and Methods

### 2.1. Study Sites

The study was conducted across two different locations in South-East France, namely the Ouveze-Ventoux and the Crau areas (Figure 1). These study sites are representative of the Mediterranean with a strong diversity of cropping systems, including fruit orchards (cherries, plums, peaches, and apricots), olives groves, and vineyards.

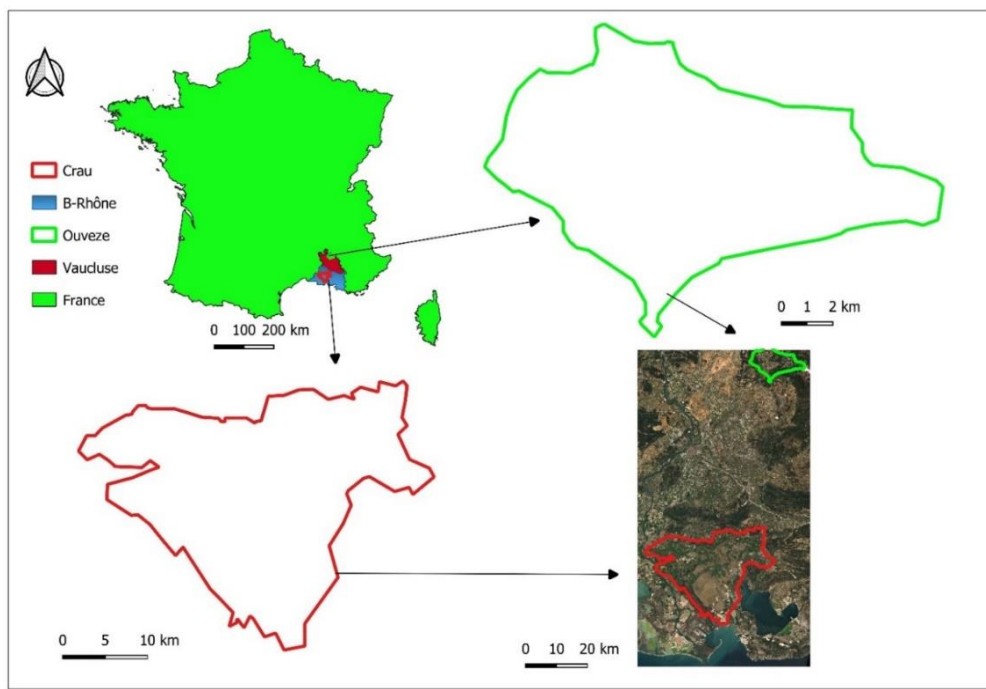

**Figure 1.** Map of France depicting the two selected study sites (Ouveze-Ventoux and Crau).

The Ouveze-Ventoux is located at 44°10′N and 5°16′E (with the lowest and highest elevation of 230 and 630 m a.s.l) in the South France of the Provence region, with a surface area of 59 km$^2$ and forest and semi-natural environments inhabiting about 57.7% [38], associated with specific bioclimatic and geomorphological context. It has the typical Mediterranean climate, identified by cold and moist winters and dry and hot summers. Annual precipitation is about 750 mm per year, with an annual mean temperature of 12.6 °C. The number of plots on Ouveze-Ventoux is about 3500, of which OC occupies about 40% (1413) and VY occupies about 34% (1186) of the cultivated area.

The Crau is positioned between 43°38′N and 5°00′E (5 m a.s.l) close to the Rhône delta in south-eastern France, with a surface area of 600 km$^2$ and a typical Mediterranean climate [10]. The annual average rainfall is 600 mm. Potential evapotranspiration of 1100 mm and mean air temperature of 14.8 °C [9,39,40]. The soils are shallow, ranging from 60–80 cm, and have 90% stones, making their water retention capacity very low. Soils irrigated via flooding methods have a loamy surface soil layer from the constantly deposited sediments, leading to a layer depth of about 60 cm, depending on the length of the irrigation period [9]. The water used for flooding irrigation contributes to more than 75% of the groundwater table, which is used for irrigation of intensive orchards and market garden productions, domestic purposes, and industrial purposes for about 280,000 people in the southern part of the area [39,40]. The number of plots in Crau is about 17,980, of which OC occupies about 11% (2050), VY occupies about 4% (790), and OL occupies about 5% (1050) of the cultivated area.

### 2.2. Ground Truth Information

The collection of ground-truth data was conducted in the two study areas during the 2016–2021 period. Plot boundaries were drawn at both sites, starting from the cadastral

survey and the RPG (Régistre Parcellaire Graphique), which is used for subsidy allocation to farmers. The boundaries were fine-tuned using an aerial picture to isolate homogeneously managed surfaces. The whole area was therefore segmented with 17,980 and 3501 plots in the Crau and Ouveze-Ventoux areas, respectively (but 16,680 and 1601 plots were used in this study due to the exclusion of 20 m from each plot boundary to avoid the effect of mixed pixels at the plot border). A subset of the controlled plot was surveyed and identified, and crop types were taken into account during field visits. In the Ouveze-Ventoux study area, a total of 234 plots (Figure 2) were identified as OC and other classes (DC), which encompass field crops, dry grasslands, and greenhouses. In the Crau study area, a total of 243 (out of 18,058) plots were selected, of which OC, (35), (60), and DC (60) encompass greenhouse, dry grass, forest, field crop, irrigated grassland, and wetland. Aerial photographs from IGN (the French national mapping service) and Google Earth images collected during the 2016–2021 period were used to understand anomalies found in vegetation time series derived from satellite, principally to assess surface heterogeneity and change in management between field visits. The number of plots in each class is given in Table 1. Fields were split into two groups dedicated to training and validation. As there are very few olive plots, the OL class was not considered at the Ouveze Ventoux site.

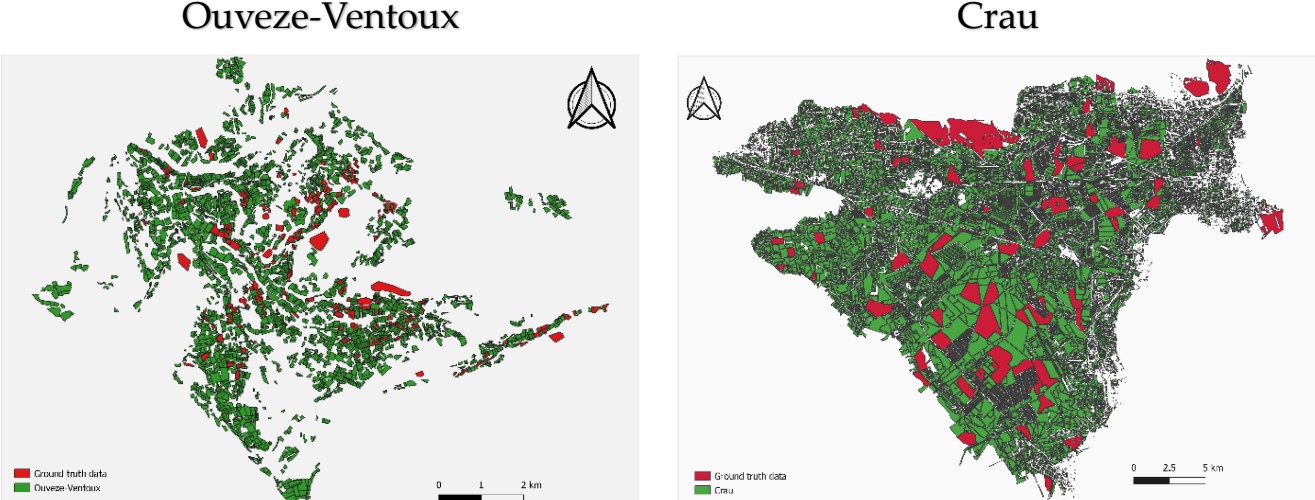

**Figure 2.** Map of the two study areas displaying locations of the selected ground-truth plot.

**Table 1.** Ground-truth information of the two study sites used for model calibration and validation during 2016–2021.

| Ouveze Ventoux Site | | Crau Site | |
|---|---|---|---|
| **Land Use** | **Ground Data (Number of Plots)** | **Land Use** | **Ground Data (Number of Plots)** |
| OC | 60 | OC | 88 |
| VY | 100 | VY | 35 |
| OL | - | OL | 60 |
| DC | 74 | DC | 60 |

### 2.3. Satellite Data

In our study, time series of Sentinel-2 (S2) optical images were utilized and obtained from both Sentinel-2A and Sentinel-2B for all dates in a given year within the 2016–2021 period, considering the visible (B2, B3, and B4), near-infrared (B8), and mid-infrared (B11 and B12) bands. We utilized the open-source service center to obtain images (https://www.theia-land.fr/, accessed on 17 May 2022); it offers cloud treatments (cloud mask) to

eliminate pixels influenced by clouds (images with >30% cloud cover), and for this obvious reason, the number of images utilized varies across study sites and years. Since Sentinel-2B satellites were functional in 2017, lesser dates were obtained from the 2016–2017 year. The number of cloud-free images (with <30% cloud cover and subsequently masked) used for each year across the two study sites is reported in Table 2. An additional cloud filter was added when creating the time series for each plot. The dates for which there was at least one pixel in the considered plot impacted by clouds were removed. This led, for a given site, to time series with different dates from one plot to another.

**Table 2.** Number of cloud-free available images across the two study sites used for the classification.

| S/N | Year | Ouveze-Ventoux | Crau |
| --- | --- | --- | --- |
| 1 | 2016 | 39 | 43 |
| 2 | 2017 | 45 | 49 |
| 3 | 2018 | 52 | 55 |
| 4 | 2019 | 51 | 56 |
| 5 | 2020 | 49 | 52 |
| 6 | 2021 | 50 | 51 |

*2.4. Vegetation Indices and Biophysical Variables*

To begin with, various vegetation indices and biophysical variables were utilized for spectral-temporal analysis, which was subdivided into those highlighting greenness, such as the popular normalized difference vegetation index (NDVI), enhanced vegetation index (EVI), and green chlorophyll vegetation index (GCVI), those highlighting moisture, such as the normalized difference moisture index (NDMI), land surface water index (LSWI), and finally biophysical variables such as leaf area index (LAI), fraction vegetation cover (FCOVER), and the fraction of absorbed photosynthetically active radiation (FAPAR), as summarized in Table 3. The biophysical variables used in this study were computed with the BVNET algorithm by utilizing the B2, B3, B4, and B8 bands. The algorithm is robust and has been fused into the S2 toolbox developed by the European Space Agency; it operates on the principles of a neural network calibrated (trained) on simulated spectral reflectance utilizing a radiative transfer model [41] and a time series of LAI implemented across every 10 m spatial resolution. In each plot polygon across the two study sites, a buffer of 20 m was removed to avoid the impact of mixed pixels at the plot boundary. The plot mean was computed by averaging the vegetation indices of all pixels in a given buffered polygon using the zonal statistics function in R [42], which were the values taken for the land classification.

**Table 3.** List of tested vegetation indices and biophysical variables.

| Full Name | Index | Formula | Reference |
| --- | --- | --- | --- |
| **Canopy greenness-related vegetation indices** | | | |
| Normalized difference vegetation index | NDVI | $\frac{NIR - RED}{NIR + RED}$ | [43] |
| Green normalized difference vegetation index | GNDVI | $\frac{NIR - GREEN}{NIR + GREEN}$ | [44] |
| Enhanced vegetation index | EVI | $2.5 * \frac{NIR - RED}{NIR + C1 * RED - C2 * BLUE + L}$ | [45] |
| Transformed soil adjusted vegetation index | TSAVI | $a * \frac{NIR - a * RED - b}{RED + a * (RED + a(NIR - b) + c * (1 + a^2)}$ | [46] |
| Atmospherically resistant vegetation index | ARVI | $\frac{NIR - (RED - 1 * (BLUE - RED))}{NIR + (RED - 1 * (BLUE - RED))}$ | [47] |
| Green chlorophyll vegetation index | GCVI | $\frac{NIR}{GREEN} - 1$ | [48] |

**Table 3.** *Cont.*

| Full Name | Index | Formula | Reference |
|---|---|---|---|
| **Water-related vegetation indices** | | | |
| Normalized difference moisture index | NDMI | $\frac{NIR - SWIR12}{NIR + SWIR12}$ | [49] |
| Land surface water index | LSWI | $\frac{NIR - SWIR1}{NIR + SWIR1}$ | [50] |
| **Biophysical variables** | | | |
| Leaf area index | LAI | | [41] |
| Fraction vegetation cover | FCOVER | | [41] |
| Fraction of absorbed photosynthetically active radiation | FAPAR | | [41] |

*2.5. Time Series Metric Derivation for Classification*

In our study, vegetation indices time series were fitted to an analytical model that represents the development of plants in relation to their phenology and uses the parameters of such relationships (PM) as a classifier used in the land classification. This is a significant variation from conventional classifiers, which target directly vegetation indices or diffusion of surface reflectance. The double sigmoid fitting function shown in Equation (1) [51,52] was fitted to the raw vegetation index time series using a non-linear least squares method (nls function in R).

$$V(t) = vmin + vamp \left( \frac{1}{1 + e^{m1 - n1t1}} - \frac{1}{1 + e^{m2 - n2t2}} \right) \tag{1}$$

where $V(t)$ stands for a given vegetation index at time $t$, $vmin$, and $vamp$ are minimum (background greenness) and amplitude parameters of one year, respectively, $m_1$, $n_1$, $m_2$, and $n_2$ are parameters controlling the curve shape (Figure 3). Some critical points are important to highlight as $t_1 = m_1/n_1$, which is the inflection point within the growth period, while $t_2 = m_2/n_2$ corresponds to the inflection point at the end of the season during the leaf senescence phase. Quantities $t_1$ and $t_2$ can be used as proxies of the start of the season (SOS) and end of the season (EOS), respectively [53]. Parameters $n_1$ and $n_2$ reflect the slope at the inflection points, $t_1$, and $t_2$.

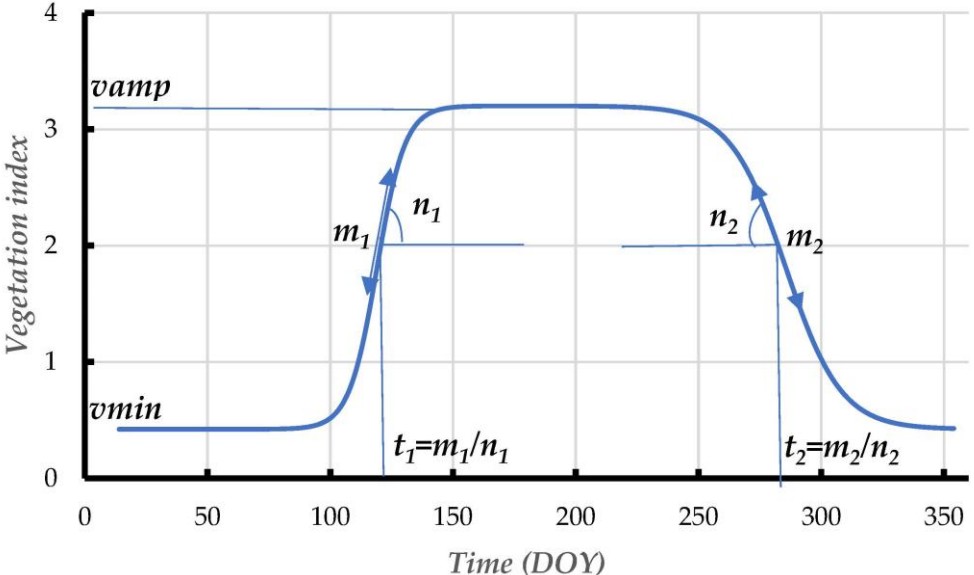

**Figure 3.** Double logistic fitting showing SOS and EOS.

With deciduous trees and annual crops, $t_1$ occurs when leaves are growing, while $t_2$ corresponds to leaf senescence. It is expected that the parameters involved in Equation (1) or their derivatives are specific to a given crop and thus can be used in calibration schemes. Furthermore, since $t_1$ is strained by the whole structure of the phenology, it is rarely impacted by noise, while $t_2$ is more undetermined for trees since defoliation is slow and relies on water accessibility and weather conditions [54]. The PM used in this study includes all these parameters ($v_{amp}$, $v_{min}$, $m_1$, $n_1$, $m_2$, $n_2$, $t_1$, $t_2$) plus the residual standard deviation (*std*) characterizing the difference between the fitted curves and the data. Some fitting examples are given in Figure 4. It shows that whatever the temporal dynamic of the vegetation, Equation (1) can be calibrated. However, when the curve does not follow the expected double logistic shape, as with a mowed grassland (Figure 4a), the *std* is high, and the phenology timing given by $t_1$ and $t_2$ ($t_1 = 200$ and $t_2 = 330$) is significantly different in the presented case from that of an orchard or a vineyard. We expected that those parameters might be considered by the classification algorithm and thus lead to the field being classified as DC.

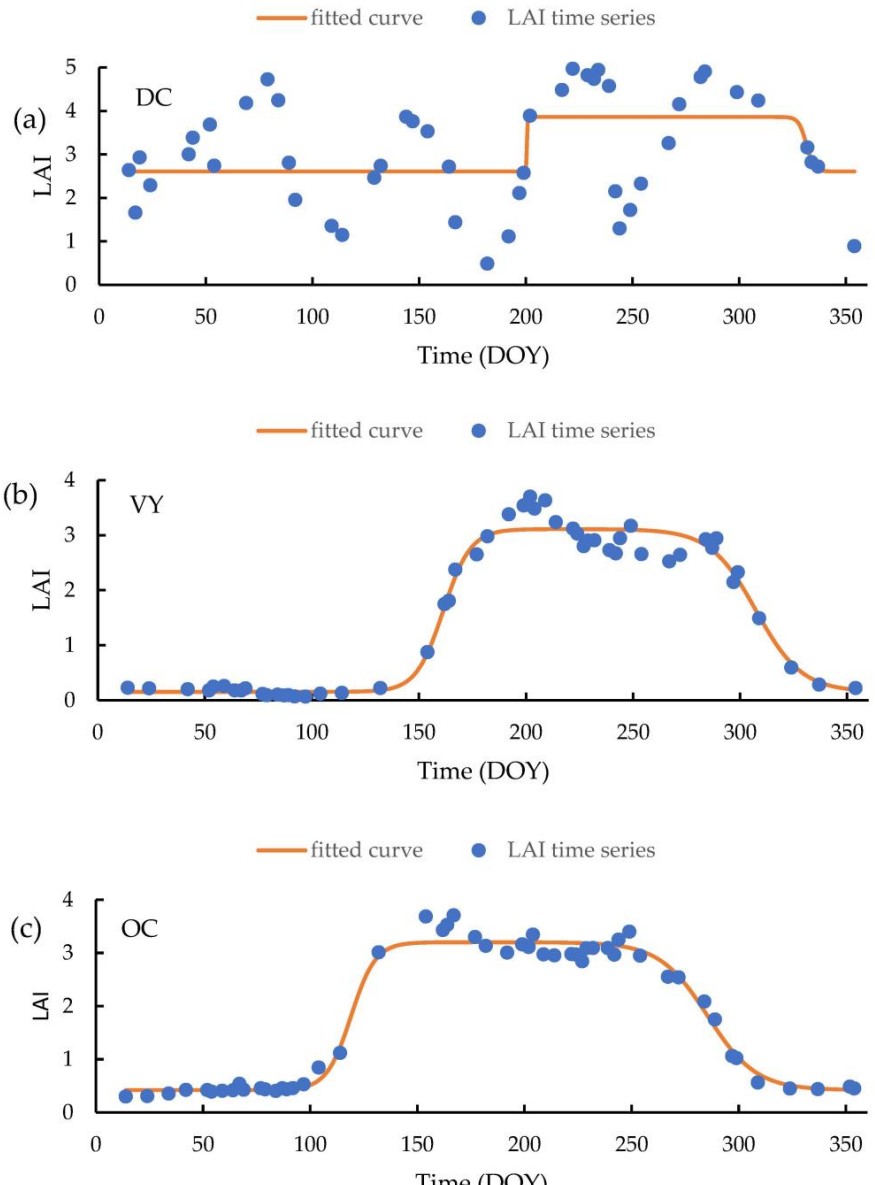

**Figure 4.** DC class with the fitted curve (**a**), VY class with the fitted curve (**b**), and OC class with the fitted curve (**c**).

*2.6. Classification Method*

Land use classification was made using a machine learning (ML) approach. Among the ML approaches, random forest (RF) is often used for land use classification. The approach is based on decision trees that can handle a lot of variables [55,56], which was the case in this study [56]. The RF method is a non-parametric ML approach that displays good results when compared to conventional parametric approaches [37]. We optimized the performance of the RF model by tuning (automatically) two significant parameters, namely *mtry* (which indicates the number of predictors tested at each tree node) and *ntree* (which displays the number of decision tree runs at each iteration). The accuracy of the classification was enhanced by tuning the number of *ntree* (after starting with the default value of 500 trees) using hyperparameter tuning for each year and each site. According to the site, the justification for such tuning is that each year has its own specific features. Thus, we adopted the value that leads to the best performance of the classifier. Finally, the model output is decided by the number of a majority of votes by the classifier ensembles.

Regarding remote sensing and land surface phenology mapping, the RF classifier remains an effective approach [35]. Thus, for classification accuracy assessments, ground-truth information was equally split into two batches for each class, namely calibration (50% proportional distribution from each class in the target population) and evaluation (50% from each crop class) datasets, utilizing a spatial cross-validation method from the CAST package in R [57]. The aforementioned spatial cross-validation aid ensures the selected ground-truth data from a similar field will be apportioned either in the calibration or evaluation dataset to keep away from over-fitting. Accuracy evaluations were performed by the confusion matrix, which gives the number of plots well classified on the diagonal and the number of erroneous detections between classes outside the diagonal, with predicted class in column and actual class in line [58]. Accuracy metrics for the classification results include overall accuracy (OA), Kohen's Kappa, which removes the chance factor, user's accuracy (UA), and producers' accuracy (PA). These metrics were computed directly from classification routines or using the CARET package in R [58] when the classification was performed in different steps.

## 3. Results

*3.1. Analysis of Temporal Profile for Orchard, Vineyard, and Olive Trees to Derive Spectral and Phenology Metrics*

In both areas, OC and VY trees are deciduous and therefore exhibit a similar temporal pattern that is characterized by a maximum plateau in the summer. However, despite the close resemblance in the temporal patterns, there are still some significant features that can be used to separate them. For instance, OC trees mostly have SOS during 60–80 DOY and 100–120 DOY time intervals in the Crau and Ouveze-Ventoux areas, respectively. With VY, such intervals are delayed by about 30 days in both areas (Figure 5). Moreover, the growth rate in vineyards is more gradual. The differences between the areas are explained by the type of orchards and the climate, with the Ouveze area being located more to the north with higher altitudes and thus lower temperatures, which induce delays in phenology. The level of the plateau is variable. It depends on the age and density of the stands. However, in general, the values obtained in summer by the VY remain lower than those of the OC, except for the irrigated VY dedicated to table grapes. The LAI variations in mid-season can be variable according to inter-row management and pruning practices. For a given field, the temporal features remain rather stable between years, meaning that the classification algorithms might be applied over different years.

OL groves are observable in the Crau. The fact of having evergreen leaves leads to a very different temporal signature in comparison to that of OC and VY, with variations rather governed by the soil cover. For the other surfaces (DC class), there is a great diversity of temporal signatures. For many of them, very different evolutions are observed (market gardening, irrigated meadows, dry meadows) from the previous cases, while ambigui-

ties could appear with some surfaces, such as wetlands, which also show a seasonality comparable to OC plots.

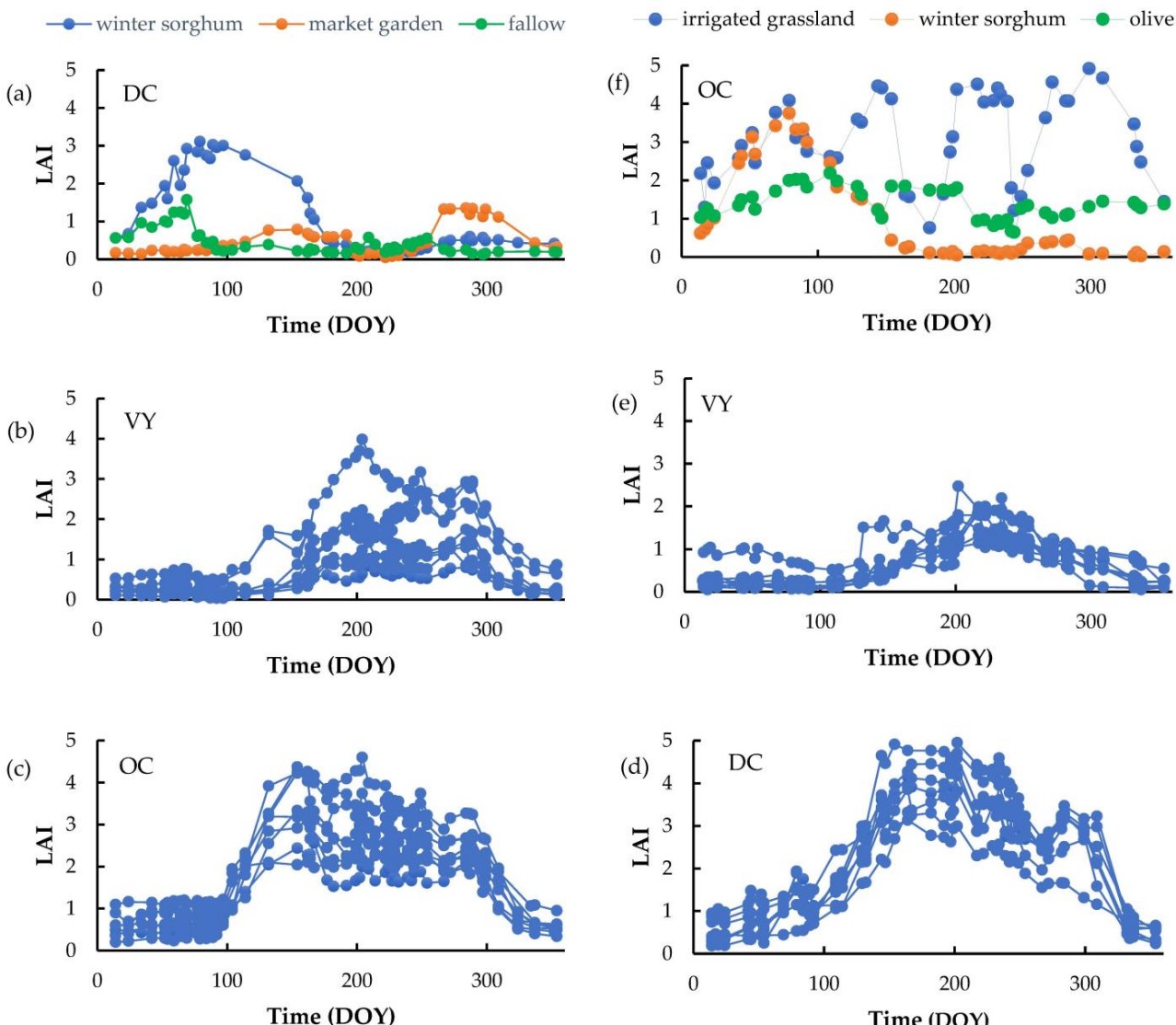

**Figure 5.** Temporal profile of DC (**a**), VY (**b**), OC (**c**) in Ouveze-Ventoux and temporal profile of DC (**d**), VY (**e**) and OC (**f**) in Crau site.

### 3.2. Selection of Indices and Biophysical Variables Using the Proposed Method

Different vegetation indices related to canopy greenness, water, and biophysical variables were analyzed to make the selection of the best indices and biophysical variables for the year 2021 at the Ouveze-Ventoux site. For all indices and biophysical variables, we fitted the double logistic model to infer the PM. These PM, together with statistical parameters qualifying the quality of the fit (*std*), were used as input in the RF algorithm to separate three classes: OC, VY, and DC. If all the considered variables representing vegetation development led to comparable results, the best results were obtained with LAI (Table 4). Moreover, using such a quantity is an advantage since it is directly comparable to field observations, which facilitate its interpretation and thus the establishment of thresholds that could be useful to conduct the classification [10]. Additionally, a further comparison was conducted from 2020 to 2018 (years with larger data acquisition dates when both Sentinel-2 satellites were operated) among the best two performing vegetation indices (NDVI and TSAVI) and LAI. This is to ensure that the LAI remains a good choice

across the years. This was the case with a K ranging between 0.85 and 0.88 when using the LAI, while the kappa ranged between 0.72 and 0.79 with the NDVI and between 0.68 and 0.74 with the TSAVI.

**Table 4.** Classification performance of different vegetation indices and biophysical variables in Ouveze-Ventoux for 2021.

| Vegetation Indices | OA(%) | K |
|---|---|---|
| NDVI | 82 | 0.71 |
| GNDVI | 80 | 0.70 |
| EVI | 82 | 0.75 |
| TSAVI | 87 | 0.85 |
| ARVI | 81 | 0.77 |
| GCVI | 73 | 0.70 |
| NDMI | 82 | 0.71 |
| LSWI | 80 | 0.77 |
| **Biophysical Variables** | **OA(%)** | **K** |
| LAI | 92 | 0.89 |
| FAPAR | 90 | 0.88 |
| FCOVER | 87 | 0.85 |

*3.3. Accuracy Assessments*

3.3.1. Delineation of Orchards and Vineyards in Ouveze-Ventoux Site

The classification at the Ouveze-Ventoux site was carried out by considering three classes, namely OC, VY, and DC. The classification was made using the PM derived from the LAI temporal profiles (Table 5). Misclassified fields mainly came from confusion between VY and DC. In about half of the cases, the confusion came from young stands having low vegetative development and thus a canopy signal that was not very clear, as displayed in Figure 6. Therefore, we merged the young stands, presented a maximum LAI lower than 0.5 to the DC class, and replayed the classification. By applying such a threshold, the classification accuracy was slightly improved (Table 6). The producer's accuracy is revealing errors due to commission, with the OC class being the best with a producer's accuracy of 0.96. To test the classification over time, the classification was performed each year from 2016 to 2021, with results summarized in Table 7. The performance of the classification was slightly affected in 2017 and 2016, and the probable explanation for this might be ascribed to fewer acquisitions of S2 images since only one of the Sentinel-2 constellations was operated.

**Table 5.** Confusion matrix for OC and VY classification using PM of LAI in 2021 (subscripts a and p correspond to actual and predicted class, respectively).

| | $DC_p$ | $OC_p$ | $VY_p$ | Total | User's Accuracy |
|---|---|---|---|---|---|
| $DC_a$ | 35 | 0 | 2 | 37 | 0.94 |
| $OC_a$ | 1 | 29 | 0 | 30 | 0.96 |
| $VY_a$ | 5 | 1 | 44 | 50 | 0.88 |
| Total | 41 | 30 | 46 | 117 | |
| Producers's accuracy | 0.86 | 0.96 | 0.96 | | |
| OA = 0.92; K = 0.89 | | | | | |

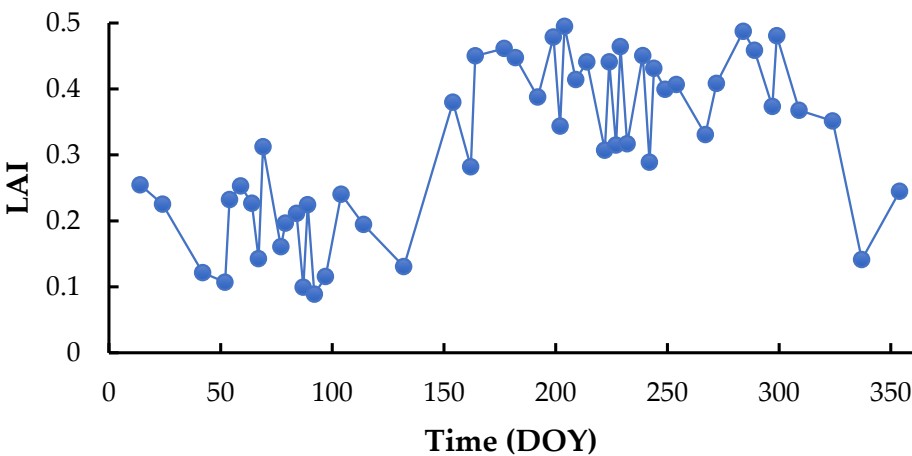

**Figure 6.** Temporal profile of a young VY misclassified as DC in Ouveze-Ventoux.

**Table 6.** Confusion matrix for OC and VY classification using PM from LAI 2021 time series and applying LAI = 0.5 for OC and VY classes (subscripts a and p correspond to actual and predicted class, respectively).

|  | $DC_p$ | $OC_p$ | $VY_p$ | Total | User's Accuracy |
|---|---|---|---|---|---|
| $DC_a$ | 37 | 0 | 2 | 39 | 0.95 |
| $OC_a$ | 0 | 30 | 0 | 30 | 1.00 |
| $VY_a$ | 1 | 1 | 46 | 48 | 0.97 |
| Total | 38 | 31 | 48 | 117 | |
| Producers's accuracy | 0.95 | 0.96 | 0.95 | | |
| OA = 0.96; K = 0.91 | | | | | |

**Table 7.** Results of OC and VY classification in Ouveze-Ventoux from 2016–201 based on PM derived from LAI time series and applying LAI = 0.5 thresholds for OC and VY classes.

| Year | Site | Accuracy Assessments | |
|---|---|---|---|
| | | OA (%) | Kappa |
| 2016 | Ouveze-Ventoux | 89 | 0.86 |
| 2017 | | 90 | 0.89 |
| 2018 | | 91 | 0.90 |
| 2019 | | 94 | 0.92 |
| 2020 | | 95 | 0.93 |
| 2021 | | 96 | 0.91 |

The classification was then applied to all fields large enough to have at least one pixel after applying a buffer of 20 m on the plot boundary. The field distribution confirms the importance of OC and VY in the Ouveze-Ventoux area, as shown in Figure 7 below.

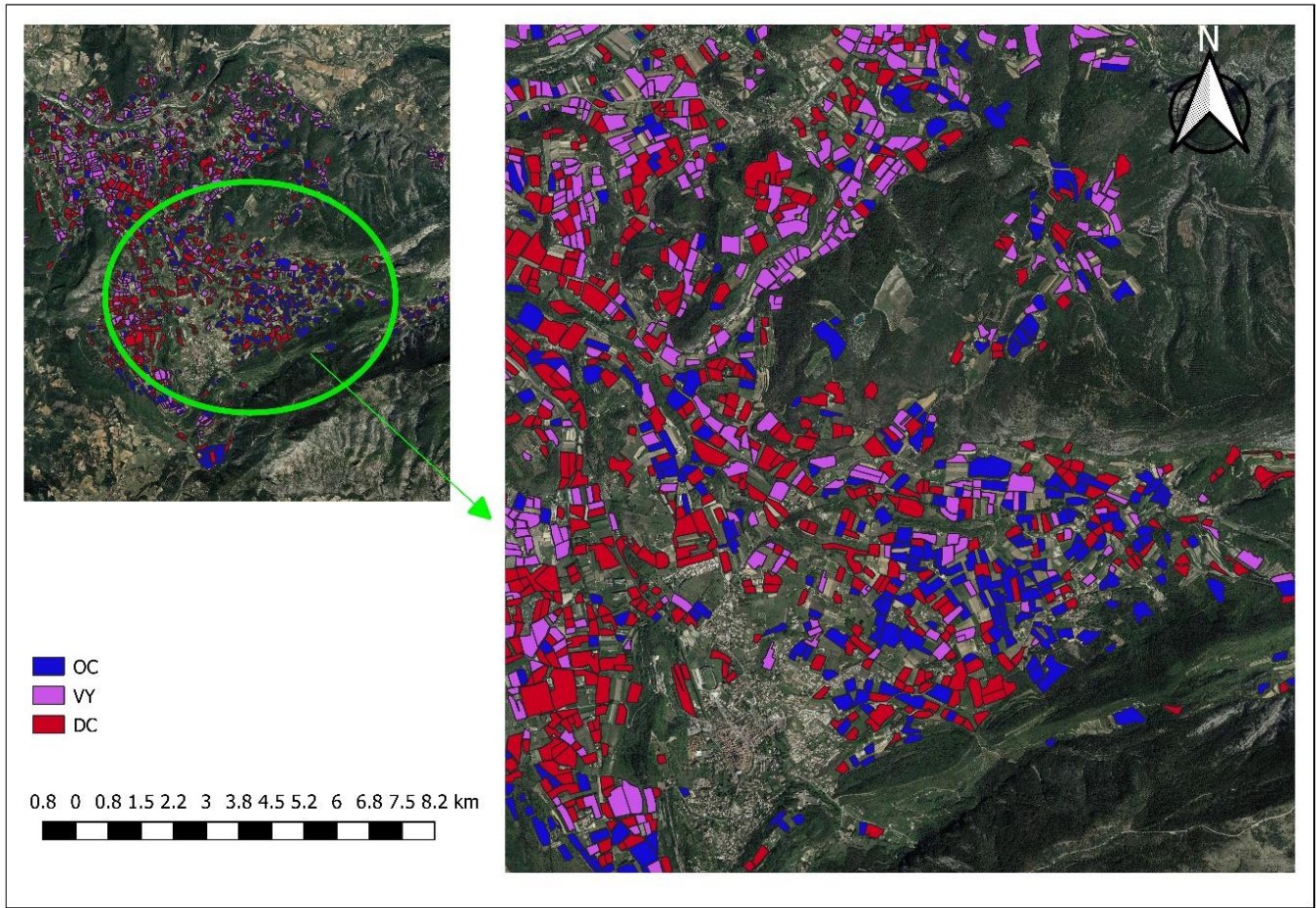

**Figure 7.** Spatial distribution of OC, VY, and DC classes in the Ouveze-Ventoux site for the year 2021.

3.3.2. Delineation of Orchards, Vineyards, and Olives in the Crau Site

On the Crau site, we find the three classes mapped on the Ouveze-Ventoux site (VY, OC, and DC), to which an OL class has been added because of the high representation of olive groves. It is also worth noting the high diversity of the DC class, with market gardening, steppe areas, wetlands, and field crops. First, we begin by conducting the classification, considering the four classes (DC, OC, OL, and VY). The results displayed in Table 8 exhibit rather weak results, with particular difficulties in delineating DC and OL classes. We decided to perform the classification in two steps. In the first step, we gathered both OL and DC in a single DC class. Then, in a second step, we delineated DC and OL. The results of the first step are reported in Table 9, showing a significant improvement with a Kappa rising from 0.69 to 0.91. Remarkably, results were also strongly improved for the VY class, with user accuracy increasing from 0.28 to 0.61. Note that the OC class was very well characterized in spite of a large diversity of tree types and varieties. As for the Ouveze-Ventoux site, an analysis of the misclassified fields showed again the difficulties in identifying the phenology in juvenile tree stands. An LAI threshold was therefore applied to the OC and VY by considering that all plots having a maximum LAI lower than 1 belong to the DC class. The quality of the classification continues to improve (Kappa = 0.95), but at the cost of not identifying young orchards and vineyards (Table 10).

**Table 8.** Confusion matrix for OC, VY, OL, and DC classification based on PM from 2021 time series (subscripts a and p correspond to actual and predicted class, respectively).

|  | DC$_p$ | OC$_p$ | OL$_p$ | VY$_p$ | Total | User's Accuracy |
|---|---|---|---|---|---|---|
| DC$_a$ | 19 | 0 | 11 | 0 | 30 | 0.63 |
| OC$_a$ | 0 | 42 | 0 | 2 | 44 | 0.95 |
| OL$_a$ | 10 | 0 | 20 | 0 | 30 | 0.67 |
| VY$_a$ | 11 | 2 | 0 | 5 | 18 | 0.28 |
| Total | 40 | 44 | 31 | 7 | 122 | |
| Producers's accuracy | 0.50 | 0.95 | 0.65 | 0.71 | | |
| OA = 0.70; K = 0.69 | | | | | | |

**Table 9.** Confusion matrix for OC and VY classification based on PM from 2021 LAI time series and after gathering OC and DC in common DC class (subscripts a and p correspond to actual and predicted class, respectively).

|  | DC$_p$ | OC$_p$ | VY$_p$ | Total | User's Accuracy |
|---|---|---|---|---|---|
| DC$_a$ | 60 | 0 | 0 | 60 | 1.00 |
| OC$_a$ | 0 | 44 | 0 | 44 | 1.00 |
| VY$_a$ | 6 | 1 | 11 | 18 | 0.61 |
| Total | 66 | 45 | 11 | 122 | |
| Producers's accuracy | 0.91 | 0.98 | 1.00 | | |
| OA = 0.94; K = 0.91 | | | | | |

**Table 10.** Same as Table 9, with an additional threshold of LAI = 1 for OC and VY classes of LAI in 2021 (subscripts a and p correspond to actual and predicted class, respectively).

|  | DC$_p$ | OC$_p$ | VY$_p$ | Total | User's Accuracy |
|---|---|---|---|---|---|
| DC$_a$ | 62 | 0 | 0 | 62 | 1.00 |
| OC$_a$ | 0 | 44 | 0 | 44 | 1.00 |
| VY$_a$ | 3 | 1 | 12 | 16 | 0.75 |
| Total | 65 | 45 | 12 | 122 | |
| Producers's accuracy | 0.95 | 0.98 | 1.00 | | |
| OA = 0.97; K = 0.95 | | | | | |

In step 2, we distinguished the olive trees (OL) in class DC. It is necessary to identify in the time series discriminating features of the olive trees, which could then be used as classifiers. In Figure 8, we show the time series of a sample of OL plots and plots identified as DC. We conducted this comparison for two variables, the LAI used in the previous classification and the GCVI, which leads to a typical signature of olive trees with a systematic decrease in the signal in the summer period compared to earlier and later periods in the year. This signature is typical of the OL class. In order to appreciate the generality of this behavior, we represented in a diagram (Figure 9) the average value of the vegetation index (VI) at the beginning of the year (between DOY 1 and 100) in abscissa and the average value in the middle of the year (between DOY 150 and 250) in ordinates. When the vegetation of the OL trees is well developed, there is a systematic decrease in the GCVI, which is all the stronger when the GCVI is high. It is also interesting to note that the area of the diagram of points covered by OL plots is specific, with little coverage of DC plots, whose dispersion in the diagram reflects the diversity of vegetation cover encountered

in this class. Similar results are obtained with LAI (Figure 9a), but with less specificity of olive plots and a less clear relationship between the difference in LAI over the two periods and the development of olive trees, as shown by the scatter of the O points. Concerning the behavior of the GCVI with the OL trees, the reasons for the reflectance ratio in the NIR and green band cannot be explained by the seasonality of the grass cover under the canopy. Indeed, the impact of the herbaceous cover should decrease with tree cover, which is contrary to what we observe. The reason could come from the orientation of the leaves and their spectral properties, which could be influenced by heat and summer water stress.

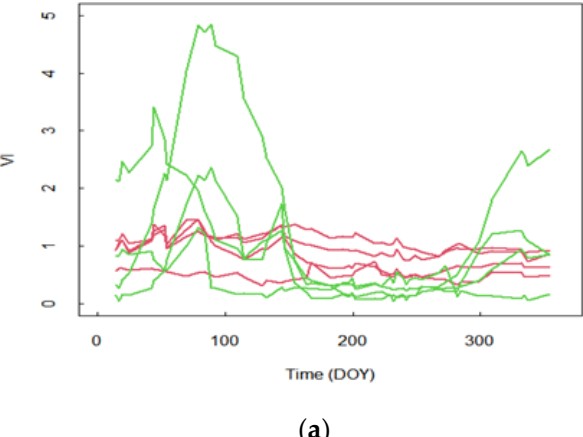

(**a**)

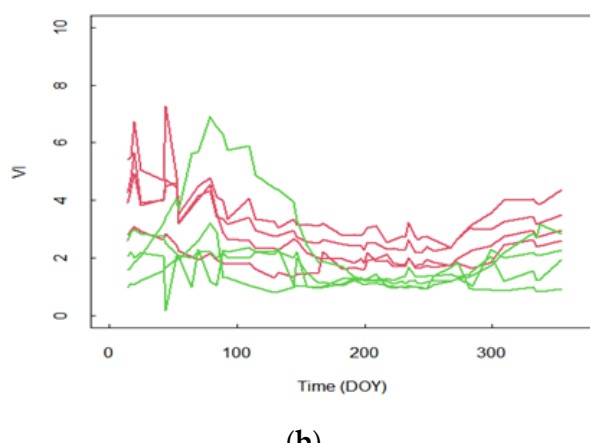

(**b**)

**Figure 8.** Vegetation indices time series observed on OL plots (red) and DC (green). In (**a**), the vegetation indices are the LAI; in (**b**), the vegetation indices are GCVI.

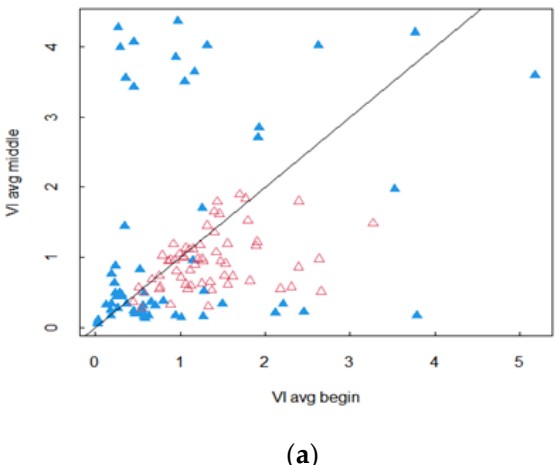

(**a**)

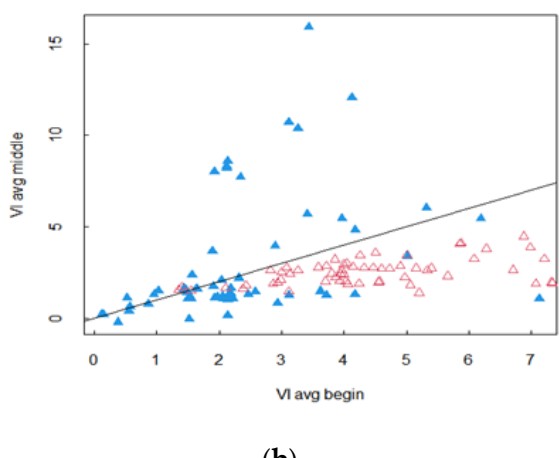

(**b**)

**Figure 9.** Average vegetation indices during the summer period (DOY 150–250) as a function of the average vegetation indices at the beginning of the year (DOY 1–100) for olive plots (red triangle) and end DC plots (blue triangle). In (**a**), the vegetation indices are the LAI; in (**b**), the vegetation indices are GCVI.

To carry out the classification, we calculated characteristics for each of the periods, namely the beginning of the year from DOY = 1 to DOY = 100 and the middle of the year from DOY = 150 to DOY = 250. The following characteristics were considered: the mean, the standard deviation, the slope, and the origin of the linear regression between the vegetation indices and time, as well as the corresponding correlation coefficient. The classification was performed with the RF method, using the 12 variables thus obtained (6 per period) as classifiers. The results are given in Table 11 when using the GCVI as the vegetation indices.

**Table 11.** Confusion matrix for DC and OL classification based on a temporal feature from 2021 GCVI time series (subscripts a and p correspond to actual and predicted class, respectively).

|  | $DC_p$ | $OL_p$ | Total | User's Accuracy |
|---|---|---|---|---|
| $DC_a$ | 25 | 5 | 30 | 0.87 |
| $OL_a$ | 1 | 29 | 30 | 0.97 |
| Total | 26 | 34 | 60 | |
| Producers's accuracy | 0.96 | 0.88 | | |
| OA = 0.91; K = 0.82 | | | | |

When using LAI instead of the GCVI, the results were degraded, with an OA = 0.82 and a Kappa = 0.70. In the misclassified plot analysis, it can be seen that the DC plots classified as OL correspond to mowed grasslands or more or less dense forests. Concerning the grasslands, it is easy to identify them because, on the periods used to calculate the characteristics of the signal, we have a strong variability of the GCVI. As far as the forests are concerned, they do not generally show the summer decrease in the GCVI. These features could not be identified by the calibration of the classification model but can easily be taken into account by post-processing. Thus, we propose to classify as DC the plots classified as OL when (1) the sum of the standard deviations of the signal obtained on each period is higher than 2, which never happens with OL trees, or (2) when the GCVI is higher than 3 and the average increases between periods 1 and 2, contrary to the behavior of OL trees. Applying this post-processing, we obtain an OA = 0.97 and a K = 0.94 (Table 12).

**Table 12.** Results of OL classification in Crau from 2016–2021 using GCVI.

| Year | Site | Accuracy Assessments | |
|---|---|---|---|
| | | OA (%) | Kappa |
| 2016 | Crau | 90 | 0.86 |
| 2017 | | 90 | 0.88 |
| 2018 | | 95 | 0.93 |
| 2019 | | 93 | 0.92 |
| 2020 | | 94 | 0.91 |
| 2021 | | 97 | 0.94 |

The final classification obtained after chaining steps 1 and 2 is displayed in Table 13, with the VY class being the most difficult to determine. The good results were maintained across the years (Table 14), with, however, some degradation for the 2016 and 2017 years, when fewer S2 data were available. One can note a small loss in accuracy in 2018, 2019, and 2020. This might be the result of the final thresholding that is specific to 2021. However, they always remain better than the results obtained when no threshold was applied.

**Table 13.** Confusion matrix of DC, OC, OL and, VY classification in Crau for 2021.

|  | $DC_p$ | $OC_p$ | $OL_p$ | $VY_p$ | Total | User's Accuracy |
|---|---|---|---|---|---|---|
| $DC_a$ | 31 | 0 | 1 | 0 | 32 | 0.97 |
| $OC_a$ | 0 | 44 | 0 | 0 | 44 | 1.00 |
| $OL_a$ | 1 | 0 | 29 | 0 | 30 | 0.97 |
| $VY_a$ | 3 | 1 | 0 | 12 | 16 | 0.75 |
| Total | 35 | 45 | 30 | 12 | 122 | |
| Producers's accuracy | 0.89 | 0.98 | 0.97 | 1.00 | | |
| OA = 0.96; K = 0.95 | | | | | | |

**Table 14.** Results of OC, VY, OL global classification accuracy in Crau from 2016–2021.

| Year | Site | Accuracy Assessments | |
|---|---|---|---|
| | | OA (%) | Kappa |
| 2016 | Crau | 89 | 0.87 |
| 2017 | | 90 | 0.87 |
| 2018 | | 92 | 0.90 |
| 2019 | | 94 | 0.91 |
| 2020 | | 93 | 0.91 |
| 2021 | | 96 | 0.95 |

*3.4. Feature Importance Ranking*

This feature importance process in the RF algorithm involves building a classification and regression tree to create OOB (out-of-bag) sample data. According to the OOB data, the RF algorithm can confirm the significant (importance) role of the input data and generate each feature's important score, which is displayed as MDA (mean decrease accuracy). The principle is to convert a feature's value to a random number, compute its influence on the model's precision, and quantify the parameter's importance according to the MDA value generated from several computations. When the value is higher, the importance variable also becomes higher [59]. Both in Ouveze-Ventoux and Crau $t_1$ and $v_{amp}$ are the two most significant feature variables, showing the importance of the start of the season and the amplitude of variation of the LAI signal (Figure 10). On the contrary, the EOS, as reflected by $t_2$, had a lower impact on the classification.

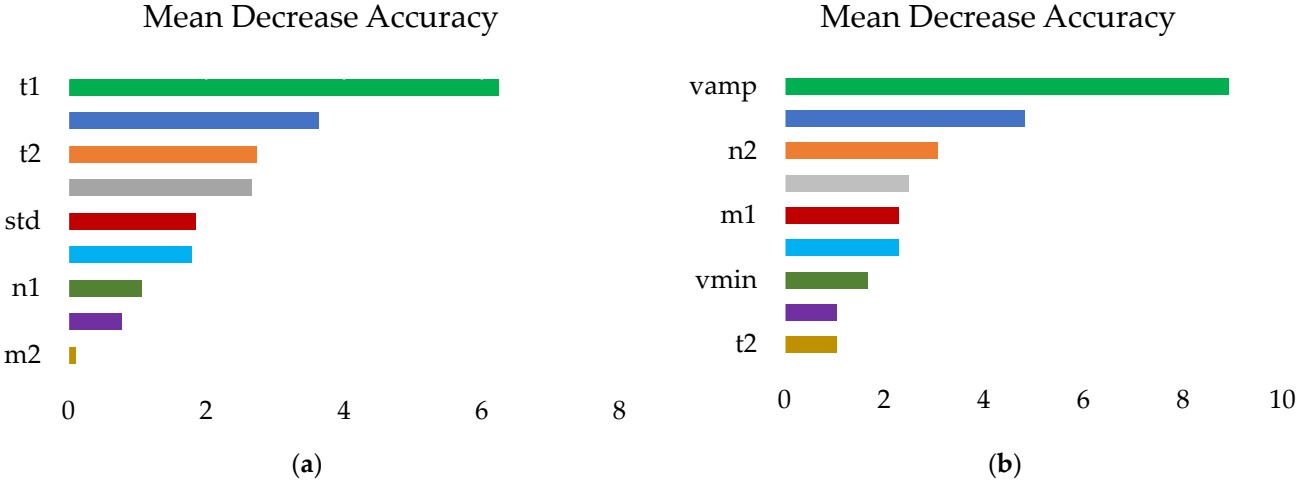

(**a**)
(**b**)

**Figure 10.** Feature importance ranking (mean decrease accuracy) for Ouveze-Ventoux (**a**) and Crau (**b**).

## 4. Discussion

*4.1. Benchmark and Novelty of the Proposed Classification Approach*

One of the reasons behind the scarcity of RS-based maps delineating fruit trees could be ascribed to the difficulty of differentiating several tree crop classes spectrally and temporarily to generate precise maps. For instance, the use of the NDVI temporal profile of fruit trees such as grapes, mango, and banana displayed no clear distinction between the different fruit tree types [60]. The novelty in our approach was to apply the classification to plant phenological traits rather than using a temporal series of images. If we obtain good results, it is necessary to assess the adding value of our approach compared to existing approaches. Two benchmarks were considered:

- A classification made directly on the times series of LAI without interpreting the phenology in time series. This was performed using the RF method. In the Crau area, we did not consider the last step separating olive orchards from other surfaces and focused on the first step which considers the following three classes only: OC, VY, and DC.
- The THEIA land use map is implemented yearly across the entire French national territory (https://www.theia-land.fr/ceslist/ces-occupationdes-sols/, accessed on 17 May 2022). It uses RF-supervised classification on all S2 dates (VIS and NIR bands) with other supplementary data such as urban maps, topography, the Corine Land Cover map, and *'Registre Parcellaire Graphique'* (RPG) that collect yearly farmer's declarations on subsidy collection from the European Union (EU). The number of identified classes was 17, of which OC (tagged as 221 and 14 in 2017 and 2018, respectively) and VY (tagged as 222 and 15 in 2017 and 2018, respectively) are inclusive, and in our study, the detection of orchards and vineyards was assessed. The maps were prepared at 10 m pixels of S2 and aggregated at the field level to be comparable with the results of our study.

The accuracies of the classification made on temporal series of LAI images were far below those obtained with our method, with an OA ranging from 41 to 60 and a Kappa ranging from 0.31 to 0.52 across the two study sites from 2016–2021 (Table 15), while we obtained a Kappa larger than 0.80. Results from 2016–2017 had the worst performance, and this might be because S2 had data acquisition limited to S2A. The difficulty of classifying LAI images directly can be explained by the very strong diversity of situations, in particular in the DC class, that might hamper the possibility of capturing specific features of the crop of interest. Better results were found with the THEIA product (Table 16), which involves much more information layers that better constrain the classification. However, the use of our method led to a significant improvement.

**Table 15.** Classification performance of the raw spectral satellite images (LAI) across the two study sites for OC and VY using RF classifier.

| Year | Site | OA (%) | K |
|------|------|--------|------|
| 2016 | Ouveze-Ventoux | 43 | 0.30 |
| 2017 | | 47 | 0.32 |
| 2018 | | 55 | 0.41 |
| 2019 | | 51 | 0.49 |
| 2020 | | 55 | 0.51 |
| 2021 | | 52 | 0.50 |
| 2016 | Crau | 41 | 0.32 |
| 2017 | | 45 | 0.31 |
| 2018 | | 49 | 0.41 |
| 2019 | | 55 | 0.52 |
| 2020 | | 60 | 0.51 |
| 2021 | | 58 | 0.49 |

**Table 16.** THEIA classification performance across the two study sites for OC and VY using RF classifier and other supplementary data.

| Year | Site | OA (%) | Kappa |
|------|------|--------|-------|
| 2016 | Ouveze-Ventoux | 73 | 0.70 |
| 2017 | | 77 | 0.72 |
| 2018 | | 75 | 0.71 |
| 2019 | | 78 | 0.75 |

**Table 16.** *Cont.*

| Year | Site | OA (%) | Kappa |
|------|------|--------|-------|
| 2020 |      | 75 | 0.71 |
| 2021 |      | 72 | 0.69 |
| 2016 | Crau | 76 | 0.73 |
| 2017 |      | 75 | 0.69 |
| 2018 |      | 79 | 0.75 |
| 2019 |      | 75 | 0.72 |
| 2020 |      | 73 | 0.68 |
| 2021 |      | 78 | 0.75 |

*4.2. Training Sample Size and Generality of the Proposed Approach across Years and Sites*

Former studies highlighted that large data samples bolster RF classification accuracy. In some situations, larger data samples might lower the RF classification accuracy, depending on the quality of the dataset. In our study, we obtained good accuracy using a small training data set of 117 (out of 1601 in Ouveze-Ventoux) and 122 (out of 16,680 in Crau). This corroborates the conclusions of Nguyen et al. [27] and Colditz [61], who found that datasets consisting of 0.15% to 0.35% of the study sites are sufficient to attain precise land cover delineations. As a rule of thumb, studies related to land use/land cover should operate with restricted data because of the excessive price of organizing field data. Therefore, to reduce the burden of ground-truth collection, we can imagine applying the RF model determined for a given year to the other years. This idea is supported by the results displayed in Figure 11, which show that the main temporal patterns of the LAI time series are rather stable from one year to another. Therefore, we have applied the RF model established in 2021 to the PM computed for the 2016, 2017, 2018, 2019, and 2020 LAI time series. The results of the classifications thus made are given in Table 17. The results revealed that the PM used for the training of the model is robust irrespective of the different years, but with slightly lower accuracies, with an average loss of 8% (max 14%) in OA and 0.10 (max 0.16) in kappa. Results obtained in the Crau are found to be a bit better, while the smallest time series in 2016 and to a lesser extent in 2017 did not have a strong impact on the classification performance. In all cases, the accuracy remains much better than that obtained when the classification was made directly on the LAI time series and comparable to the THEIA classification.

**Table 17.** Classification performance using PM for classifying orchards and vineyards applying the predicted model of 2021 across years and sites.

| | Accuracy Assessment | | | | |
|---|---|---|---|---|---|
| | | Calibrated RF Model | | 2021 Model across Years | |
| Year | Site | OA | Kappa | OA | Kappa |
| 2016 | Ouveze-Ventoux | 0.89 | 0.86 | 81 | 0.72 |
| 2017 |      | 0.90 | 0.89 | 83 | 0.76 |
| 2018 |      | 0.91 | 0.90 | 82 | 0.79 |
| 2019 |      | 0.94 | 0.92 | 83 | 0.81 |
| 2020 |      | 0.95 | 0.93 | 86 | 0.85 |

**Table 17.** *Cont.*

| | | Accuracy Assessment | | | |
|---|---|---|---|---|---|
| | | **Calibrated RF Model** | | **2021 Model across Years** | |
| **Year** | **Site** | **OA** | **Kappa** | **OA** | **Kappa** |
| 2016 | Crau | 89 | 0.87 | 83 | 0.79 |
| 2017 | | 90 | 0.87 | 82 | 0.78 |
| 2018 | | 92 | 0.90 | 85 | 0.80 |
| 2019 | | 94 | 0.91 | 87 | 0.83 |
| 2020 | | 93 | 0.91 | 86 | 0.80 |
| | | **Calibrated RF model** | | **2021 model across sites** | |
| 2021 | Ouveze Ventoux | 96 | 0.91 | 71 | 0.65 |
| 2021 | Crau | 89 | 0.87 | 60 | 0.51 |

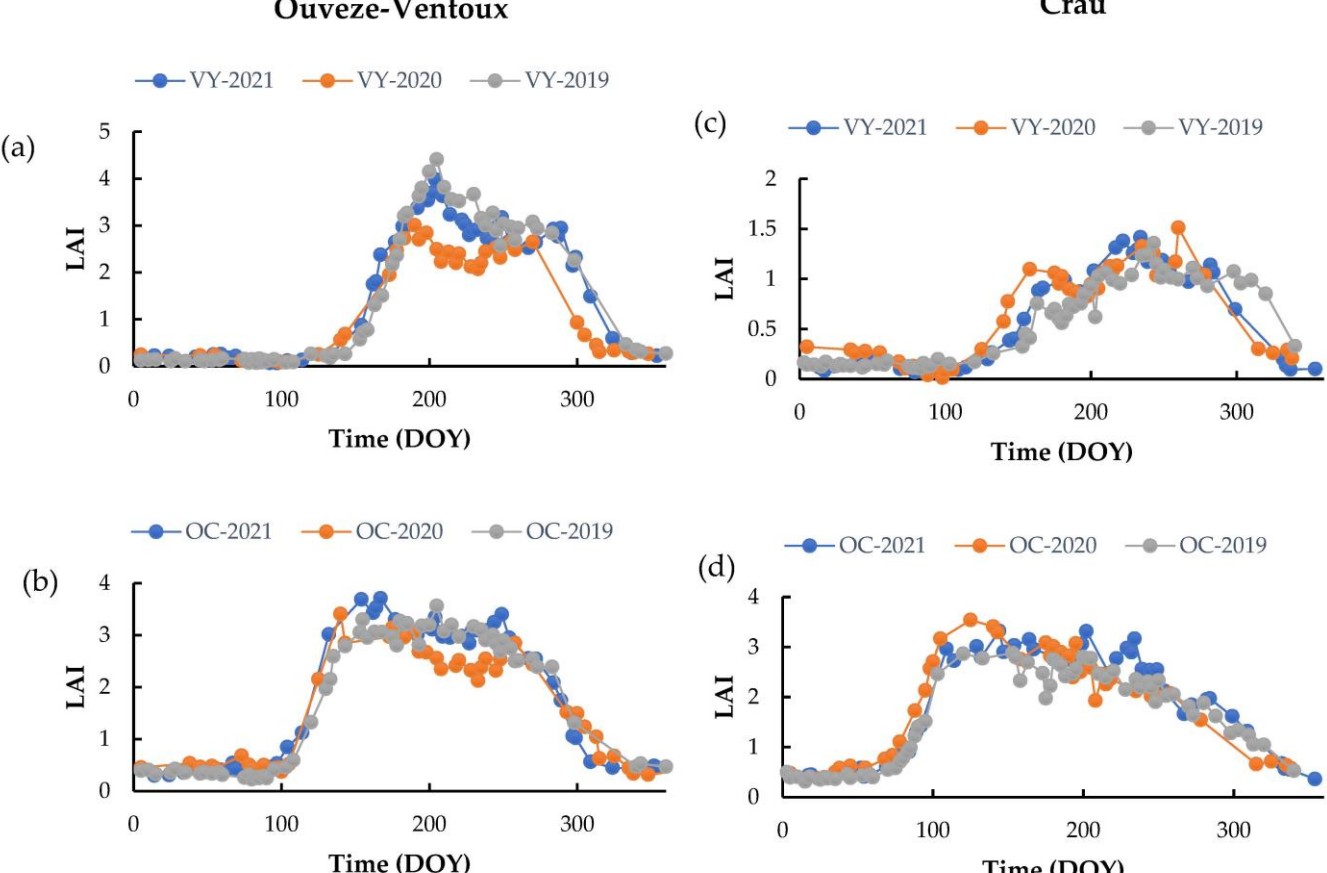

**Figure 11.** Temporal profile of OC and VY in Ouveze-Ventoux (**a**,**b**) and Crau (**c**,**d**) study areas from 2019 to 2021.

The same exercise was performed to make comparisons across sites (i.e., the model of 2021 established in the Ouveze-Ventoux site to be used for Crau and vice versa). In this exercise, the performance of the model was affected (Table 17). This further corroborates the fact that each model generated is strictly adapted to a given location; the climatic and geographical variations and diversity in crop management practices across the two sites might be responsible for the decline in results since the model was adapted to a different location (based on their PM).

### 4.3. Limitations and Prospects of the Proposed Classification Approach

Despite the novelty and good performance of our proposed approach, we are faced with quite a few drawbacks, which are to be highlighted in this section. One of the obvious drawbacks is the inability of the approach to successfully classify young OC and VY. In general, the canopy size of young fruit trees (OC and VY) is very scanty (low) and consequently creates room for misclassification since our approach is based on the temporal dynamics of the LAI. One of the main reasons is the contribution of inter-row vegetation, which can be dominant in young stands. As a result, we can end up with an LAI signal that is no longer dominated by the plant of interest. In Figure 12, we see that the SOS is earlier with the VY plot than with the OC plot, which is contrary to the expected result. Such drawbacks are minimized by considering only OC and VY plots with sufficient development, which led us to put thresholds in our reference dataset by assimilating several young OC and VY to the DC class. To obtain an optimal result, this thresholding depends on the considered area, which is probably a reflection of different cultural practices for the management of the inter-row. The determination of these thresholds is a real limitation of the method. There is therefore a strong stake in working on the separation of the contributions coming from the inter-row vegetation and the canopy of the stand of interest. A finer analysis of the spectral signature and/or a finer analysis of the temporal dynamics could contribute to a better separation of these two components and thus work on the specific signal of the crop of interest.

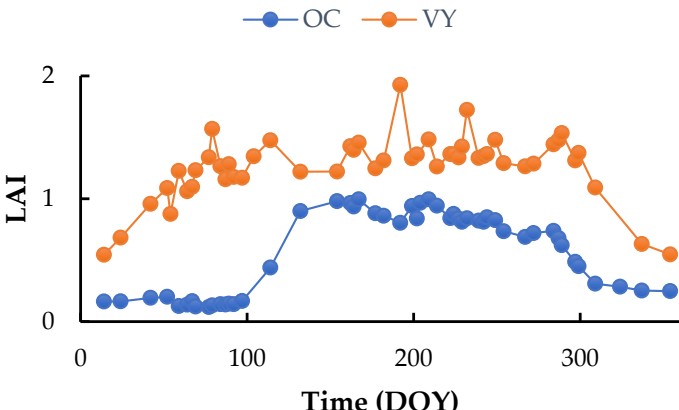

**Figure 12.** Temporal pattern of a young OC field and a temporal pattern of a young VY.

Another instance is the inability of the approach to accurately classify heterogeneous plots. The trees are sparse due to age differences, creating so many sources of interference, most especially from soil background, among others. This creates sources for interference with the main canopy of the heterogenous orchard or vineyard fields since only the mean of all the total pixels in a given farm plot is utilized, thereby creating a different form of the temporal pattern from that of the fruit trees of interest. Such deviations from the main temporal patterns of fields used for training the model consequently lead to degraded fits by the model and misclassifications when making predictions to a global extent.

Despite the highlighted drawbacks mentioned above, the prospect of PM-based delineations is encouraging for many reasons. Our analysis of the data showed that we can separate three (3) crucial fruit tree types (OC, VY, and OL). PM-based classifiers were used only as input classifiers to the RF model, so the approach can be extended to profit from some spectral bands and some particular vegetation indices, such as the enhanced bloom index (EBI) [62]), capable of separating the individual orchard (such as cherry, plum, apricot, peach, nectarine, etc.) and vineyard (table and wine) classes. Despite our work using S2, we might be faced with issues of missing data (gap) in some areas due to the presence of the cloud affecting the capacity of inferring accurate PM and thus affecting the classification accuracy. Future satellite systems with better spatial and temporal resolutions

can be merged with S2, or even synergy between optical and synthetic aperture radar (SAR) can be exploited.

## 5. Conclusions

Fruit tree delineations have been a difficult topic in crop delineation using remote sensing information. S2 has offered an encouraging avenue to build a classification strategy based on crop phenology and the temporal features of canopy development. Therefore, our study proposed a novel method to identify deciduous and evergreen fruit trees, such as OC, VY, and OL, by using a time series of LAI (for OC and VY) and GCVI (for OL) derived from S2 data to infer PM as classifiers used by an RF algorithm. The method has been developed and implemented in two areas (Ouveze-Ventoux and Crau) located in the south-east of France, separated by 100 kilometers. The main differences are the climate, with a cooler and wetter climate in the Ouveze-Ventoux area, and the composition of the DC class, which is strongly different between sites. The obtained performances led to an overall accuracy ranging between 0.89 and 0.96 and a Kappa index ranging between 0.87 and 0.95. This is far better than the results we can obtain by applying the RF method to LAI time series (the same used to infer the phenology metrics) and significantly better than the THEIA classification, which is an operational tool implemented over the French territory using multiple sources of ground information. Moreover, as the method is independent of the satellite acquisition dates, we can apply an RF classification model obtained from one year to the next while maintaining reasonable accuracy.

While this study shows the value of using phenology and leaf development parameters to identify perennial woody crops, the use of phenology may have some limitations. It is shown that the differences in phenology induced by the climate do not allow the use of a calibrated RF model from one site to another. The proposed generic approach must therefore be calibrated for each study area as soon as a temporal shift in phenology is expected. Moreover, in the case of a mixed cover composed of plants with different temporal dynamics, it may be difficult to capture the phenology of the plant of interest. This is the case in this study, with young plantations having an inter-row with grass. Mixing the signals from the tree canopy with those from the inter-row does not allow the identification of the phenological traits of the trees. To overcome such limitations, additional information, such as that provided by textural analysis of remote sensing images, might be an interesting avenue to improve the results. From that perspective, the use of satellites with different resolutions can be envisaged.

**Author Contributions:** Conceptualization, M.A.A. and A.C.; data curation, M.A.A., F.F. and G.P.; formal analysis, M.A.A.; methodology, M.A.A., A.C. and G.P.; software, G.P.; supervision A.C. and D.C.; writing—original draft, M.A.A. and A.C.; writing—review and editing, M.A.A., A.C. and D.C. All authors have read and agreed to the published version of the manuscript.

**Funding:** This work was funded by Petroleum Technology Development Fund (PTDF) under the Federal Ministry of Petroleum, Nigeria, in collaboration with INRAE-EMMAH Avignon as part of a PhD research program.

**Data Availability Statement:** Sentinel-2 data are available at the following hyperlink: https://www.theia-land.fr/, accessed on 17 May 2022. THEIA classification data can be found at the following hyperlink: https://www.theia-land.fr/ceslist/ces-occupation-des-sols/, accessed on 17 May 2022. Inglada, Jordi, Vincent, Arthur, & Thierion, Vincent. (2018). Theia OSO Land Cover Map 2018 [Data set]. Zenodo. https://doi.org/10.5281/zenodo.3613415 (accessed on 17 May 2022). Plot boundary map and evaluation data over the CRAU and OUVEZE-VENTOUX are available on request from the corresponding author.

**Acknowledgments:** The authors would like to thank the reviewers for helping us improve the quality of this research work.

**Conflicts of Interest:** We are not aware of any conflict of interest linked to this publication, and there is no significant financial aid for this work that could have influenced its outcome.

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
