# Peer review of "Delineation of Orchard, Vineyard, and Olive Trees Based on Phenology Metrics Derived from Time Series of Sentinel-2"

_remotesensing, doi:10.3390/rs15092420_

Round 1

Reviewer 1 Report

The authors propose a fast, accurate and cost-effective analytical approach for the delineation of fruit orchards (OC), vineyards (VY), and olive groves (OL) in the Mediterranean (Southern France) considering two locations about 100 km apart but with different climatic conditions and plant cover other than the desired perennial woody crops.

Authors must follow the specifications of the Microsoft Word template or LaTeX template to prepare their manuscript. Abstract: A single paragraph of about 200 words maximum.

In the introduction, highlights are missing to attracting the attention of the reader for novelty and relevance of the proposed contribution.

Conclusions could explore experimental results in a more consistent approach and give an assertive response to the proposed objectives.

Author Response

We thank the reviewer for its wize and helpfull comment. The comments were adressed as follow :

Authors must follow the specifications of the Microsoft Word template or LaTeX template to prepare their manuscript. Abstract: A single paragraph of about 200 words maximum.

Response : the abstract has been adjusted to a maximum words of 200 and specification of the template followed

In the introduction, highlights are missing to attracting the attention of the reader for novelty and relevance of the proposed contribution.

The introduction was improved. First we reduce the first part to make the introduction more focussed on the paper objectif which the delineation of perennial woody crop. In the last paragraph of the introduction we try to better justify our approach in comparison to the highlight of the bibliography analysis.

Conclusions could explore experimental results in a more consistent approach and give an assertive response to the proposed objectives.

We are not sure to fully understand what is expected here. However, the conclusions was improved by removing speculative statements and complement the conclusions with limitations in using PM to implement the classification.

Reviewer 2 Report

This passage describes a study aimed at developing a classification approach to accurately and rapidly identify fruit orchards, vineyards, and olive groves in Southern France using remote sensing data from the Sentinel-2 mission. The study notes that while previous research has focused on herbaceous and gramineous crops, identifying woody crops like fruit trees and vines is challenging due to their management practices and variability. The approach developed in this study uses phenology metrics, derived from fitting a double logistic model to temporal vegetation index profiles, to classify crops. The random forest algorithm was used to identify woody crops using the generated phenology metrics. The best results were obtained with the leaf area index, while the temporal features of the green chlorophyll vegetation index were found to be most appropriate for delineating olive groves in the DC class. The approach achieved an overall accuracy ranging from 89-96% and Kappa of 0.86-0.95 by considering each study site and year separately. The study concludes that using phenological traits rather than raw time series data can improve accuracy and reduce the need for ground truth information. However, the study also cautions that classification accuracy may be affected by site-specific factors like climatic conditions, cultivars, and management practices.

the paper has some strengths and weaknesses.

Strengths:

The paper proposes a rapid, accurate, and cost-effective analytical approach for the delineation of fruit orchards, vineyards, and olive groves in Southern France, which can be useful for agricultural monitoring.

The approach uses phenology metrics derived from temporal Sentinel-2 time series, which allows for the identification of woody crops that are difficult to distinguish from other land covers.

The method was tested on different vegetation indices, and the results show an overall accuracy ranging from 89-96% and Kappa of 0.86-0.95, which is a good performance.

Weaknesses:

The paper acknowledges that fewer results have been obtained on woody crops, which suggests that more research is needed in this area.

The paper mentions that using a classification model calibrated in one site and applied to another led to a strong degradation of the classification accuracy, which limits the generalizability of the approach.

The paper does not provide information on the potential limitations of using phenology metrics or the implications of relying on remote sensing data for agricultural monitoring.

Author Response

We thank the reviewer for his wize and helpful comments. We take his comment as follow.

This passage describes a study aimed at developing a classification approach to accurately and rapidly identify fruit orchards, vineyards, and olive groves in Southern France using remote sensing data from the Sentinel-2 mission. The study notes that while previous research has focused on herbaceous and gramineous crops, identifying woody crops like fruit trees and vines is challenging due to their management practices and variability. The approach developed in this study uses phenology metrics, derived from fitting a double logistic model to temporal vegetation index profiles, to classify crops. The random forest algorithm was used to identify woody crops using the generated phenology metrics. The best results were obtained with the leaf area index, while the temporal features of the green chlorophyll vegetation index were found to be most appropriate for delineating olive groves in the DC class. The approach achieved an overall accuracy ranging from 89-96% and Kappa of 0.86-0.95 by considering each study site and year separately. The study concludes that using phenological traits rather than raw time series data can improve accuracy and reduce the need for ground truth information. However, the study also cautions that classification accuracy may be affected by site-specific factors like climatic conditions, cultivars, and management practices.

the paper has some strengths and weaknesses.

Strengths:

The paper proposes a rapid, accurate, and cost-effective analytical approach for the delineation of fruit orchards, vineyards, and olive groves in Southern France, which can be useful for agricultural monitoring.

The approach uses phenology metrics derived from temporal Sentinel-2 time series, which allows for the identification of woody crops that are difficult to distinguish from other land covers.

The method was tested on different vegetation indices, and the results show an overall accuracy ranging from 89-96% and Kappa of 0.86-0.95, which is a good performance.

Weaknesses:

The paper acknowledges that fewer results have been obtained on woody crops, which suggests that more research is needed in this area.

The paper mentions that using a classification model calibrated in one site and applied to another led to a strong degradation of the classification accuracy, which limits the generalizability of the approach.

Answer : The method is generic and can be applied in different areas but needs a local calibration as soon as phenology shift is expected. This was said more clearly in the conclusion

The paper does not provide information on the potential limitations of using phenology metrics or the implications of relying on remote sensing data for agricultural monitoring.

Answer : Potential limitations are discussed  in the results and summarized in the improved conclusion.

Reviewer 3 Report

Thanks for your contribution. In this article, the authors are mainly focused on crop classification based on phenology metrics by using time-series Sentinel-2 data.

Here are my comments:

(1) This article is focused on crop classification based on phenology metrics, so a most important thing is how to fit the discrete vegetation index values via the double sigmoid function as shown in equation 1. However, I could not find any fitted curve from the figures shown in this manuscript. Moreover, this article lacks the detail of how to process the discrete time-series vegetation index values, did the authors filter the time-series index values?

(2) The plots in Fig. 4 are lack of standardization. For example, it is impossible to distinguish year-to-year data from Fig.4 (e) - Fig.4 (g), the range of the y-coordinate are not consistent for Fig.4 (a) and Fig.4 (d), the scale marks of the x-coordinate are incomplete, etc. Moreover, the LAI index curves varies relatively obvious from year to year, see DOY 100-300 from Fig.4 (b) and Fig.4 (c). What is the reason for this phenomenon to occur? Please comment on this.

(3) As the authors mentioned in Line 437-439, PM and residual std were used as input for RF classification to separate OC, VY, and DC. However, the DC class contains greenhouse, dry grass, field crop and so on, the time-series vegetation indices of these feature types can show very different curves and may not be fitted by equation 1, as those shown in Fig. 4(a) and Fig. 4(b). I was wondering how the authors treat with the diverse PM and std parameters of these different feature types that contained in DC class. Did the authors used the same PM and std values for these feature types? Please comment on this.

(4) From Table 4, we can determine that the LAI index superior to the other vegetation indices. However, this may not be the case for year from 2016-2020. The authors are suggested to select the optimal index based on the multi-year data.

(5) Since the RF algorithm was used for classification, I am very interested in feature importance ranking, please provide such plots and relevant descriptions.

(6) As the authors mentioned in Line 580-593: “The shape of the plot is often correlated the crop type”, so why not trying to add texture features as input to ML classifier to help distinguish crops from DC class?

Author Response

(1) This article is focused on crop classification based on phenology metrics, so a most important thing is how to fit the discrete vegetation index values via the double sigmoid function as shown in equation 1. However, I could not find any fitted curve from the figures shown in this manuscript. Moreover, this article lacks the detail of how to process the discrete time-series vegetation index values, did the authors filter the time-series index values?

Response : time series of DC, VY and OC with their curves have been provided in the materials and methods (see Figure 4). The fitting procedure is now described with more details. In fact a non linear fitting algorithms was applied to the raw time series to compute the double logistic model parameters

(2) The plots in Fig. 4 are lack of standardization. For example, it is impossible to distinguish year-to-year data from Fig.4 (e) - Fig.4 (g), the range of the y-coordinate are not consistent for Fig.4 (a) and Fig.4 (d), the scale marks of the x-coordinate are incomplete, etc. Moreover, the LAI index curves varies relatively obvious from year to year, see DOY 100-300 from Fig.4 (b) and Fig.4 (c). What is the reason for this phenomenon to occur? Please comment on this.

Response : These plots have been standardized (all the plots in this part are for year 2021 but belonging to different fields. This is mentioned clearly in the legend

(3) As the authors mentioned in Line 437-439, PM and residual std were used as input for RF classification to separate OC, VY, and DC. However, the DC class contains greenhouse, dry grass, field crop and so on, the time-series vegetation indices of these feature types can show very different curves and may not be fitted by equation 1, as those shown in Fig. 4(a) and Fig. 4(b). I was wondering how the authors treat with the diverse PM and std parameters of these different feature types that contained in DC class. Did the authors used the same PM and std values for these feature types? Please comment on this.

The double logistic was applied to all situation. In the case when the Vegetation did not follow a double logistic form, the retrieved parameter were different to that of the other class and in general led to a strong std. the Random forest algorithm is able to sort out these features and thus detect field belonging the DC class. The DC exemple shown in Figure 4a was selected to illustrate This feature and commented in the text

(4) From Table 4, we can determine that the LAI index superior to the other vegetation indices. However, this may not be the case for year from 2016-2020. The authors are suggested to select the optimal index based on the multi-year data.

Response : This is as well the case for the year 2020 to 2018 ( we consider years after 2017 when both S2A and S2B were in operation), LAI still has an edge over NDVI and TSAVI and it was commented in the body of the paper.

(5) Since the RF algorithm was used for classification, I am very interested in feature importance ranking, please provide such plots and relevant descriptions.

Response : The feature importance ranking (plots) were provided and commented in a seperate section of the text for the classification of VY and OC.

(6) As the authors mentioned in Line 580-593: “The shape of the plot is often correlated the crop type”, so why not trying to add texture features as input to ML classifier to help distinguish crops from DC class?

We removed Figure 9 adressing the shape of the plot in order to balance a bit the increase in size of the paper after addressing the comment of the reviewers. Thus don’t comment the shape of the plots anymore. We agree that adding texture feature might be an interesting avenue to improve the results. This opening was mentioned in the conclusion.

Round 2

Reviewer 1 Report

The authors have improved the manuscript significantly and have answered all the concerns of reviewers. I support its publication.

Author Response

No correction being asked

Reviewer 2 Report

None

Author Response

No correction being asked

Reviewer 3 Report

 I could not see all the plots in Figure 4, Figure 4(b) and Figure 4(c) are missing, maybe due to the eding error. Besides, what does it mean by saying "We expected that those parameted might be sorted out y the classification algorithm."?

Author Response

The problem of the figure was likely due to the sofware used to display the text. On our editor the figures were good. A pdf version is provided.

The problematic sentence was rephrased as follow :

We expected that those parameters might be considered by the classification algorithm and thus would lead to the field classified as DC.